# Identification and Functional Analysis of a Lysozyme Gene from *Coridius chinensis* (Hemiptera: Dinidoridae)

**DOI:** 10.3390/biology10040330

**Published:** 2021-04-14

**Authors:** Hai Huang, Juan Du, Shang-Wei Li, Tao Gong

**Affiliations:** Guizhou Provincial Key Laboratory for Agricultural Pest Management of Mountainous Regions, Institute of Entomology, Guizhou University, Guiyang 550025, China; hh118545@163.com (H.H.); juandudj@163.com (J.D.); gongtao458@163.com (T.G.)

**Keywords:** *Coridius chinensis*, c-type lysozyme, innate immunity, muramidase activity, antibacterial activity

## Abstract

**Simple Summary:**

As a medicinal insect, *Coridius chinensis* contains many active polypeptides. Extracts from *C. chinensis* are usually complex and it is not clear which polypeptides are effective medicinal ingredients. In addition, we also need to figure out the functions of various immune effectors in the innate immunity of *C. chinensis*. To explore the function of lysozyme in *C. chinensis*, a lysozyme gene *CcLys2* was screened and identified from the transcriptome data of *C. chinensis.* The results showed that CcLys2 had a typical domain of the c-type lysozyme, belonging to the H-branch of the c-type lysozyme. The lysozyme Cclys2 is an effective immune effector in the immune response of *C. chinensis* and can be stimulated by bacterial infection. Like typical c-type lysozyme, Cclys2 has lytic activity against Gram-positive bacteria. The research holds promise for functional annotation of similar proteins from other dinidoridae insects and provides the theoretical feasibility for the development of medicinal components in *C. chinensis.* Our results also provide data for further investigating the origin and evolution of insect lysozymes.

**Abstract:**

*Coridius chinensis* is a valuable medicinal insect resource in China. Previous studies have indicated that the antibacterial and anticancer effects of the *C. chinensis* extract mainly come from the active polypeptides. Lysozyme is an effective immune effector in insect innate immunity and usually has excellent bactericidal effects. There are two kinds of lysozymes in insects, c-type and i-type, which play an important role in innate immunity and intestinal digestion. Studying lysozyme in *C. chinensis* will be helpful to further explore the evolutionary relationship and functional differences among lysozymes of various species and to determine whether they have biological activity and medicinal value. In this study, a lysozyme CcLys2 was identified from *C. chinensis*. CcLys2 contains 223 amino acid residues, and possesses a typical domain of the c-type lysozyme and a putative catalytic site formed by two conserved residues Glu32 and Asp50. Phylogenetic analysis showed that CcLys2 belongs to the H-branch of the c-type lysozyme. The analysis of spatiotemporal expression patterns indicated that *CcLys2* was mainly expressed in the fat body of *C. chinensis* adults and was highly expressed in the second- and fifth-instar nymphs. In addition, *CcLys2* was significantly up-regulated after injecting and feeding bacteria. In the bacterial inhibition assay, it was found that CcLys2 had antibacterial activity against Gram-positive bacteria at a low pH. These results indicate that CcLys2 has muramidase activity, involves in the innate immunity of *C. chinensis,* and is also closely related to the bacterial immune defense or digestive function of the intestine.

## 1. Introduction

Lysozyme was first discovered in human tears and saliva, and was found to dissolve bacterial cell walls and to kill bacteria. A total of four lysozymes have been identified in animals, commonly designated as the c-type (chicken or conventional type), the g-type (goose type), the i-type (invertebrate type), and the ch-type (chalaropsis type) lysozymes [1,2]. At present, the main lysozymes identified are c-type and i-type lysozymes in insects. In *Drosophila melanogaster*, 18 lysozyme genes have been identified, including 5 i-type lysozymes and 13 c-type lysozymes [3,4,5,6]. In total, eight lysozymes were also found in *Anopheles gambiae* (lysozymes c1-c8) [7].

As a group of evolutionarily conserved enzymes, lysozymes mediate insect innate immunity [8]. Multicellular animals resist infection by two systems called innate immunity and acquired immunity; however, in the long-term process of biological evolution, insects have formed different immune systems from vertebrates, and their immune responses mainly depend on the innate immune system [9,10]. Innate immunity in insects is mainly divided into humoral immunity and cellular immunity; humoral immunity is the main immunity of lysozyme participation. The c-type lysozyme usually has muramidase activity and can hydrolyze the β-1,4-glycosidic bond between N-acetylmuramic acid and N-acetylglucosamine in peptidoglycan polymer, so it has an antibacterial activity [11]. The i-type lysozymes have muramidase and isopeptidase activities, but the i-type lysozymes in insects lack muramidase activity. In addition, some i-type lysozymes cannot be induced by immune stimulation, which indicates that they may have acquired new and undetermined functions in the process of evolution [12]. Researchers have suggested that c-type lysozymes are not only involved in immune defense but are also involved in insect digestion. For example, c-type lysozymes 1 and 2 from *Musca domestica* have digestive functions, which aid in the use of bacteria as a food source. These lysozymes have high expression in the gut and optimal lytic activity at a lower pH [13,14].

In general, the c-type lysozymes have been considered to have immune or digestive functions, such as *Bombyx mori* lysozyme BmLZ [15], human lysozyme HLYZ [16], and *M. domestica* lysozymes 1 and 2 mentioned above. On the other hand, the c-type lysozymes also play a biological function of non-muramidase activity. Some c-type lysozyme-like proteins without catalytic residues still have non-enzymatic antibacterial activity and can be strongly induced by immune stimulation, such as BLLP1 from *B. mori* and ALLP1 from *Antheraea mylitta* [17]. The non-bacteriolytic c-type lysozyme-like protein SLLP1 of mammals is located in the acrosomal matrix of the sperm head and is considered to play a role in sperm–egg binding [18]. The insect lysozymes also have chitinase activity, which is much higher than that of vertebrate lysozymes [6]. Previous studies have shown that the function of lysozyme in various insects is different, whether it is c-type or i-type lysozyme; therefore, it is significantly important to reveal the function of lysozyme in insects. This provides not only data support for the research of insect immune mechanisms but also a theoretical basis for the development and utilization of insect resources. There are two ways to obtain lysozymes: direct extraction such as hen egg-white lysozyme or recombinant expression by genetic engineering technology, such as c-type lysozyme Fi-Lyz of *Fenneropenaeus indicus* and SmLysC of *Scophthalmus maximus* [19,20].

*Coridius chinensis* belongs to Dinidoridae, Hemiptera, and is usually used as a medicine in China. Previous studies have revealed that the hemolymph of *C. chinensis* has broad-spectrum antibacterial activity, which mainly comes from some antimicrobial peptides (AMPs) in the hemolymph [21]. In this study, a lysozyme gene named as *CcLys2* (GenBank accession number: MN816376) was cloned based on the transcriptome data of *C. chinensis*. The relationship between CcLys2 and the c-type lysozymes of other insects was elucidated by sequence analysis and phylogenetic tree construction. The spatiotemporal expression profile of *CcLys2* was analyzed at different developmental stages and in various tissues of adults by using real-time quantitative PCR (RT-qPCR). The expression levels of *CcLys2* in adults injected or fed bacteria were analyzed, which further revealed the role of CcLys2 in the innate immunity of *C. chinensis*. *CcLys2* was successfully expressed in *E. coli* BL21 (DE3) and this enzyme was proved to have the antibacterial effect by using bacterial growth inhibition test.

## 2. Materials and Methods

### 2.1. Insects and Bacteria

The *C. chinensis* laboratory population was collected from Guiyang, Guizhou Province, China, in 2019. *C. chinensis* was kept at 28 ± 1 °C and a 75 ± 5% relative humidity under a photoperiod of 10: 14 h (L: D). Both larvae and adults feed on fresh pumpkin leaves. *Escherichia coli* BL21 (DE3) cells with 30% glycerol (Beyotime Biotech, Shanghai, China) and expression vector pET-28b(+) (Beyotime Biotech, Shanghai, China) were stored in a refrigerator at −80 °C. *E. coli* (ATCC 25922), *Pseudomonas solanacearum* (ATCC 33192), *Salmonella typhi* (CMCC 50071), *Micrococcus luteus* (CMCC 28001), *Bacillus subtilis* (CMCC 63501), and *Staphylococcus aureus* (ATCC 25923) were deposited in the Institute of Entomology, Guizhou University, China.

### 2.2. Characteristic Analyses of CcLys2

The signal peptide was analyzed by SignalP 4.1 (http://www.cbs.dtu.dk/services/SignalP, 6 July 2020). The molecular weight, isoelectric point, and hydrophilicity of CcLys2 were predicted on the platform of Expasy ProtParam tool (https://web.expasy.org/protparam, 6 July 2020). Prediction of glycosylation sites with NetOGlyc 4.0 Server (http://www.cbs.dtu.dk/services/NetOGLyc, 6 July 2020) and NetNGlyc 1.0 Server (http://www.cbs.dtu.dk/services/NetNGlyc, 6 July 2020). DiANNA 1.1 webserver (http://clavius.bc.edu/~Clotelab/DiANNA, 6 July 2020) was used to predict disulfide bond. Subcellular localization was performed using TargetP 1.1 (http://www.cbs.dtu.dk/services/TargetP, 6 July 2020). Domain prediction was performed by using SMART (http://smart.embl-heidelberg.de, 6 July 2020). The homology was analyzed by BLAST in NCBI Nr protein database and the phylogenetic tree was constructed by using the neighbor-joining method in MEGA X software with 1000 runs. The three-dimensional structure of the mature CcLys2 was predicted by SWISS-MODEL homologous modeling (https://swissmodel.expasy.org, 16 October 2020) and its molecular graphic was drawn using PyMOL 2.4 (Schrodinger, New York, NY, USA).

### 2.3. RNA Extraction and cDNA Synthesis

The HP Total RNA Kit (Omega Bio-Tek Inc., Norcross, GA, USA) was used to extract the total RNA from different developmental stages (the eggs, first-fifth instar nymphs, and females and males) and from various adult tissues (the head, integument, fat body, midgut, hemolymph, testis, ovary, and muscle). The hemolymph was extracted by double tube centrifugation [22]. Four holes were made at the bottom of the inner tube in the double tubes using a 0.1-mm diameter metal needle burned on an alcohol lamp. After the head, legs, and wings of *C. chinensis* were removed with sterile and nuclease-free scissors, the insect body was cut into pieces with a nuclease-free scalpel, and then was put into the double tubes and centrifuged at 2500× *g* for 10 min at 4 °C. The centrifuged product was transferred to a new 1.5-mL nuclease-free centrifuge tube and stored at −80 °C. The purity and concentration of RNA were determined using a NanoDrop 2000 spectrophotometer (Thermo Fisher Scientific, Waltham, MA, USA), and the quality of RNA was detected by using 1% agarose gel electrophoresis. RNA was used as a template to synthesize cDNA using a RevertAid First Strand cDNA Synthesis Kit (Thermo Fisher Scientific, Waltham, MA, USA). The synthesized cDNA was diluted to 400 ng/μL and stored at −20 °C.

### 2.4. Spatiotemporal Expression Profile of CcLys2 

The expression levels of *CcLys2* at different developmental stages and in various adult tissues were determined using CFX96 Touch Real-Time PCR (Bio-Rad, Hercules, CA, USA). The specific primers specific to *CcLys2* for RT-qPCR were designed by Primer Premier 6.0 (PREMIER Biosoft, San Francisco, CA, USA) (Table 1). RT-qPCR was carried out in a 20-μL reaction system containing 1 μL of cDNA templates, 1 μL each of forward and reverse primers, 10 μL of 2 × SYBR Select Master Mix (Thermo Fisher Scientific, Waltham, MA, USA), and 7 μL of nuclease-free water. The PCR reaction parameters were: 50 °C for 2 min; 95 °C for 2 min; and 35 cycles of 95 °C for 15 s, 58 °C for 15 s, and 72 °C for 1 min. The product was verified by dissociation curve analysis. Each sample was repeated three times. The *β-actin* gene of *C. chinensis* (GenBank accession number: MK370101) was used as the internal reference, and the primers used are shown in Table 1.

### 2.5. Analysis of Bacterial Infection

*M. luteus* and *E. coli* were inoculated in 20-mL Luria-Bertani (LB) medium and cultured in an incubator at 200 rpm at 37 °C. When the OD_600_ value of the culture reached 0.1, the bacteria were centrifuged at 10,000× *g* for 5 min, and collected and washed with phosphate-buffered saline (PBS). These two bacteria were mixed and then suspended in PBS to make OD_600_ = 0.01. One hundred healthy *C. chinensis* adults were randomly selected to be intraperitoneally injected using 1 μL of bacterial suspension. The adults were collected at 6, 8, 12, 24, 36, 48, 60, and 72 h after infection and then frozen with liquid nitrogen for standby. We also used the method of feeding bacteria to study the effect of CcLys2 in the midgut of *C. chinensis*. Inoculated in 250-mL LB medium and cultured overnight to OD_600_ = 4, *M. luteus* was centrifuged at 10,000× *g* for 5 min. The bacteria were washed with PBS and then suspended in PBS to regulate OD_600_ = 2. The fresh pumpkin leaves and stems were cut into pieces and soaked in the suspension of *M. luteus* for 30 min, and then transferred to a transparent plastic box (15 mm × 25 mm). A total of 50 *C. chinensis* adults starved for 12 h were put into the box that was sealed with a nylon net instead of the plastic cover to ensure air circulation and was put into an artificial climate box. The midguts of *C. chinensis* at 6 h after feeding bacteria were dissected and collected for quick freezing with liquid nitrogen. Healthy adult samples without any treatment were used as the control group. Total RNA extraction, cDNA synthesis, and RT-qPCR were all described above.

### 2.6. Construction of Recombinant Expression Vector

According to the preference of codon usage in *E. coli*, the nucleotide sequence of the mature CcLys2 was optimized by the GenSmar Codon Optimization Tool (Version Beta 1.0) (https://www.genscript.com/tools/gensmart-codon-optimization, 15 October 2020). The restriction endonuclease recognition sites for *Nde* I and *Eco*R I were added at the 5′ and 3′ ends of the coding region of *CcLys2*, respectively. The 6 × His-tag was added to the N-terminal of the amino acid sequence to facilitate the purification of the recombinant protein. The optimized sequence was synthesized (Beyotime Biotech, Shanghai, China) and then subcloned into the pET-28b(+) vector. The recombinant plasmid was transferred into *E. coli* BL21 (DE3) cells. The transformed *E. coli* solution was coated on the LB agar plate containing 50 μg/mL of kanamycin and incubated overnight at 37 °C. The positive clones were screened out by PCR. The universal sequencing primers (T7-F and T7-R) for the pET-28b(+) vector were selected (Table 1), and PCR was carried out with 2 × PCR Master mix (Tsingke, Beijing, China). The reaction parameters were as follows: 95 °C for 3 min; 35 cycles of 95 °C for 30 s, 55 °C for 30 s, and 72 °C for 1 min; and a final extension of 72 °C for 10 min. The positive clones were submitted to Sangon Biotech (Shanghai, China) for sequencing to confirm the correct insertion sequence.

### 2.7. Expression and Purification of Recombinant Protein

The positive clones were cultured at 37 °C overnight and then inoculated in 50-mL LB medium. When the bacterial solution grew to the appropriate concentration, IPTG was added to a final concentration of 1 mM. After induction for 4 h at 37 °C, the bacteria were harvested by centrifugation at 4000× *g* for 20 min at 4 °C, and then resuspended in PBS and lysed on ice through sonication. The precipitate and supernatant were collected after centrifugation at 10,000× *g* for 10 min. The precipitate was dissolved in denaturation buffer (pH 7.5 PBS and 8 M urea). The protein samples were analyzed by sodium dodecyl sulfate–polyacrylamide gel electrophoresis (SDS-PAGE) and then visualized by staining with Coomassie brilliant blue.

According to the above conditions, the induced expression strain was amplified to 1 L. The expressed protein with the 6 × His-tag at the N-terminus was purified under denaturing conditions using the Ni-NTA agarose (Beyotime Biotech, Shanghai, China) as instructed by the manufacturer’s protocol. The concentration of purified CcLys2 was determined by the BCA Protein Assay kit (Solarbio, Beijing, China) and 20 μL of CcLys2 was used for SDS-PAGE. The target protein was identified by Western blotting, using anti-6 × His rabbit polyclonal antibody (1: 800 dilution) as the primary antibody and horseradish peroxidase-conjugated goat anti-rabbit IgG (1: 5000 dilution) as the secondary antibody. The purification protein was refolded by dialysis against descending concentrations of urea (4, 2, 1, and 0.1 M) containing 0.4 mM oxidized glutathione and 4 mM reduced glutathione for 12 h at 4 °C. After protein renaturation, the dialysis solution was taken out and centrifuged to collect the supernatant.

### 2.8. Activity Assay of CcLys2

The muramidase activity of the recombinant CcLys2 was measured with *M. luteus* as substrate by the turbidimetric method [23,24]. Hen egg white lysozyme (HEWL) was used as the positive control (2 μg/μL), and the lyophilized powder of non-induced bacteria medium was used as the negative control (2 μg/μL). *M. luteus* was cultured to OD_450_ = 1 and then harvested by centrifugation. The bacteria were lyophilized using a Christ ALPHA 1-2 LDplus freeze dryer (Christ, Osterode, NI, Germany) and resuspended in 0.1 M potassium phosphate buffer with different pH (4.5, 5, 5.5, 6, 6.5, 7, 7.5, and 8.0), adjusting the OD_450_ value of the bacterial solution to 0.5. The 150 μL of the above bacterial suspension was mixed with 50 μL of renatured CcLys2 (2 μg/μL), and the mixtures were added into a 96-well plate, followed by incubation for 60 min at 37 °C. The absorbance at 450 nm was measured using a Multiskan GO microplate spectrophotometer (Thermo Fisher Scientific, Waltham, MA, USA).

### 2.9. Antibacterial Assay of CcLys2

The antibacterial activity of CcLys2 was determined by the agar plate diffusion method. HEWL was used as the positive control, and the culture supernatant of non-induced recombinant *E. coli* was used as the negative control. *M. luteus* was picked out with sterile toothpicks and inoculated into centrifuge tubes containing 1-mL LB medium without antibiotic. The bacteria were cultured overnight for 12 h at 200 rpm and 37 °C. The 100-mL Mueller-Hinton (MH) agar medium was sterilized at 121 °C for 30 min and then cooled to 50 °C. The 10 μL of bacterial solution was mixed with 20 mL of MH agar medium (1: 2000) and poured into a 90-mm Petrie Dish. Then, holes with a diameter of 6 mm were drilled in the agar plate using a puncher, and HEWL (10 μg), CcLys2 (10 μg), and non-induced bacterial medium (10 μg) were separately added into these holes. After standing at room temperature for 2 h, the plates were incubated at 37 °C for 24 h and photographed by using a ChemiDoc MP Imaging System (Bio-Rad, Hercules, CA, USA). The antibacterial activity of CcLys2 against five other kinds of bacteria (*E. coli*, *P. solanacearum*, *S. typhi*, *B. subtilis,* and *S. aureus*) was detected using the same method as the above.

### 2.10. Data Analysis

The expression levels of the *CcLys2* gene at different developmental stages, in various adult tissues, and in adults induced by bacteria were calculated using the 2^−ΔΔCt^ method. Data of *CcLys2* expression levels were analyzed using SPSS 22.0 statistical software (SPSS Inc., Chicago, IL, USA), and multiple comparisons were performed using one-way analysis of variance (ANOVA) and the Duncan’s multiple range test. A *P*-value less than 0.05 was considered as statistical significance.

## 3. Results

### 3.1. Characteristic Analyses of CcLys2

The CcLys2 zymoprotein is a hydrophilic protein with a molecular weight of 24.8 kDa and an isoelectric point of 5.09. CcLys2 consists of 223 amino acid (AA) residues, containing a signal peptide of 13 AAs at the N-terminus and a mature protein of 210 AAs (Figure 1a). CcLys2 is longer than the typical lysozyme in the AA sequence. Just after the domain LYZ1 (14–138) of CcLys2, there is a special sequence composed of 85 AAs (139–223), with 1 N-glycosylation site and 17 O-glycosylation sites. In general, glycosylation can increase the thermal stability, conformational stability, and solubility of protein; therefore, this sequence may play an important role in the stability of CcLys2. Interestingly, homology search revealed that this special sequence was also found in lysozymes from other dinidoridae insects (https://www.ncbi.nlm.nih.gov/, 6 January 2021). Subcellular localization prediction showed that CcLys2 located outside a cell and belonged to a secretory protein. Domain analysis indicated that CcLys2 belonged to the c-type lysozyme/α-lactalbumin family. It contains a conserved domain LYZ1 of the c-type lysozyme (Figure 1b) and eight cysteine residues that forms four disulfide bonds (C_6_-C_124_, C_27_-C_113_, C_62_-C_73_, and C_69_-C_87_). In addition, the mature peptide of CcLys2 also has catalytically essential residues, glutamic E32 and aspartic D50, which are equivalent to E33 and D50 of lysozyme BmLZ in *B. mori*, but differs from the non-catalytic lysozyme-like proteins (LLPs), ALLP1 and BLLP1; therefore, CcLys2 is likely to have catalytic activity. Homology modeling showed that the mature CcLys2 formed seven α-helices, three β-pleated sheets, and a groove for substrate binding between the helices and pleated sheets. The groove contained the conserved residues E32 and D50 on either side of it, which formed the putative catalytic site (Figure 2a,b).

### 3.2. Homologous and Phylogenetic Analyses

The deduced amino acid sequence of mature CcLys2 was compared with those of the known eight c-type lysozymes and two lysozyme-like proteins from the insect as illustrated in Figure 3. The multiple sequence alignment showed the two catalytically essential residues, glutamic and aspartic, and the completely conserved eight-cysteine residue motif. In addition, multiple sequence alignment between CcLys2 and eight other known lysozymes revealed that CcLys2 belonged to the H-branch of the c-type lysozymes that shared a characteristic histidine residue (H) which replaced the conserved tyrosine residue (Y) adjacent to the catalytic aspartate residue (D) in most known c-type lysozymes, such as BmLZ from *B. mori* and AgLYSC1 from *A. gambiae* (Figure 3).

Lysozyme-like protein CcLLP3 of *C. chinensis* cluster together with other catalytically-inactive proteins, forming a distinct sub-group of lysozyme-like proteins (Figure 4). The putative muramidases CcLys2 clustered within a further distinct group, the members of this group belong to the H-branch of c-type lysozymes (Figure 4), except for HaLys3. CcLys2 shared 74.31% similarity with that of *Halyomorpha halys* and 65.62% with the c-type lysozyme of *Plautia stali* via BLAST search against NCBI Nr protein database using AAs of the zymoprotein. The results showed that *C. chinensis* is the closest relative of *H. halys* and *P. stali*.

### 3.3. Spatiotemporal Expression Patterns

The RT-qPCR results showed that the *CcLys2* gene was differentially expressed at various developmental stages of *C. chinensis*, with the highest expression level in the second-instar nymph followed by the fifth-instar nymph and the lowest level in the first-instar nymph. The *CcLys2* expression was relatively low in eggs, first-, third-, and fourth-instar nymphs, without significant difference in expression levels between them. There was not significant difference in the *CcLys2* expression between males and females (Figure 5a). The expression level of *CcLys2* in the second-instar nymph was 1.38 times that in the fifth-instar nymph and 21.98 times that in the egg. In addition, in the eight adult tissues tested, *CcLys2* was expressed at the highest level in the fat body, followed by the hemolymph and midgut, and at the lowest level in the integument (Figure 5b). The *CcLys2* expression level in the fat body was 3.66 times that in the hemolymph and 58.94 times that in the head.

### 3.4. Expression profile of CcLys2 after Bacterial Infection

The expression level of *CcLys2* in adults was up-regulated 6 to 60 h after injecting *M. luteus* and *E. coli*, reaching the highest level at 60 h, and then declined. The *CcLys2* expression levels 8 h post injection was 3 times that without infection and its expression level 60 h post injection was 76.42 times that without infection (Figure 6a). The findings implied CcLys2 was involved in the immune response to bacterial infection. The *CcLys2* expression in the midgut 6 h post feeding with *M. luteus* was nearly three times as high as that in the control (Figure 6b). The results indicated that CcLys2 can be induced by feeding bacteria, which suggested that it was also closely related to the bacterial immune defense or digestive function of the intestinal.

### 3.5. Identification of Recombinant Strain 

After the plasmid pET−28b(+)−CcLys2 was transferred into *E. coli* BL21 (DE3), agarose gel electrophoresis of PCR products displayed a band at about 1 kb, which was consistent with the expected size (Appendix A). The sequencing result showed that the expression vector pET−28b(+)−CcLys2 was successfully constructed. SDS-PAGE showed a band at about 28 kDa, which was slightly larger than the expected molecular weight (25.6 kDa) of the recombinant CcLys2 protein (Figure 7a). After the His-tagged fusion protein was purified using Ni-NTA affinity chromatography, SDS-PAGE showed a clear band at about 28 kDa (Figure 7b), and Western blot revealed a blotting band at about 28 kDa (Figure 7c and Appendix A). These results indicate that the fusion CcLys2 protein has been successfully expressed. The concentration of fusion CcLys2 determined by the BCA method was 0.4 mg/mL.

### 3.6. Muramidase Activity of CcLys2

Using *M. luteus* as the substrate, the bacterial cells were lysed by lysozyme CcLys2, and the turbidity of bacterial liquid decreased gradually with the progress of the lytic reaction. The results revealed that the optimal pH value of CcLys2 was 6.0 (Figure 8a). The non-induced bacterial culture could not dissolve *M. luteus*, whereas the HEWL and CcLys2 could lyse this bacterium. Recombinant CcLys2 had muramidase activity, but the activity was lower than that of HEWL (Figure 8b).

### 3.7. Antimicrobial Activity of CcLys2

The antibacterial activity of the fusion CcLys2 was detected by the agar plate diffusion method. CcLys2 produced a lytic zone against *M. luteus* on the agarose plate, but did not produce a inhibition zone against *S. aureus* (Figure 9a,b). In addition, CcLys2 also showed an inhibitory effect on *B. subtilis*. Like most c-type lysozymes, CcLys2 has no inhibitory effect on tested Gram-negative bacteria such as *E. coli* (Figure 9a,b). These results indicated that CcLys2 had antibacterial activity against most Gram-positive bacteria.

## 4. Discussion

As a common protease, the c-type lysozyme is generally considered to have immune or digestive functions. Based on the transcriptome of *C. chinensis*, we screened and characterized a c-type lysozyme, CcLys2. It has a typical conservative domain of the c-type lysozyme and contains glutamate (E32) and aspartate residues (D50) at the predicted catalytic sites. In general, glutamate and aspartate are the catalytically essential residues for the c-type lysozyme to exert its muramidase activity, which has been confirmed by the structural and functional studies of HEWL and other lysozymes [20,25]. E35, as a proton donor of glucoside oxygen, facilitates the cleavage of the C-O bond, while D52 forms a glycosyl intermediate through its carboxyl group as a nucleophile [1,26]. In addition, CcLys2 also has eight conserved cysteine residues that can form four intramolecular disulfide bonds that are also crucial for lysozyme activity [27]. Conservative disulfide bonds may play a role in maintaining structural stability in CcLys2, as observed in HEWL [1]. The three-dimensional (3D) structure of CcLys2 was constructed by homologous modeling. There is a crack in the 3D structure of the c-type lysozyme, which divides the c-type lysozyme into two regions, α-helices and β-pleated sheets. The crack is usually used as a binding site to peptidoglycan, where two essential residues play a catalytic role [1]. In most known c-type lysozymes, the conserved amino acid adjacent to the catalytically essential aspartate residue (D) is tyrosine residue (Y), but it is replaced by histidine residue (H) in CcLys2. Such lysozymes have been noticed in previous studies and named as the H-branch of lysozymes [28]. Significantly, there is at least one c-type lysozyme homolog in all sequenced insect genomes or transcriptomes (https://www.ncbi.nlm.nih.gov, 6 January 2021), suggesting that the c-type lysozyme exists widely in insects. The c-type lysozymes of different insects seem to come from a common ancestor and the reason for the divergence is evolutionary selection pressure, resulting in a diverse set of homologues.

Interestingly, the amino acid sequence of CcLys2 is longer than that of the typical lysozymes. Adjacent to domain LYZ1 of CcLys2, there is a special sequence composed of 85 AAs, which contains many glycosylation sites. Generally, the thermal stability, enzymatic stability, conformational stability, and solubility of protein will significantly increased after glycosylation [29,30]. The enzyme recognition sites on the surface of glycosylated proteins will be covered by sugar chains in varying degrees, which will interfere with the recognition of protease and the enzymatic hydrolysis process, resulting in the enhanced resistance of protein to protease degradation [31]. Some c-type lysozymes exert a digestive influence on the digestive organs of insects, and are highly expressed and stably maintained in the intestine containing a large number of proteases; therefore, the special sequence may play an important role in maintaining the stability of CcLys2. In addition, studies have shown that glycosylated proteins can effectively enhance the immune response. On the one hand, glycosylated proteins can induce immune responses. The glycoprotein side chain is usually the recognition bridge between cells, receptor and ligand, and immunoglobulin and pathogen [32]. On the other hand, glycosylation can reduce the immunogenicity of proteins; therefore, glycosylated proteins are of great significance to immune response.

Two i-type lysozymes (Pclysi1 and Pclysi2) found in *Procambarus clarkii* had the activities of muramidase, isopeptidase, and chitinase [33]. The lysozyme gene from *Triatoma infestans* was also up-regulated after molting, which seems to indicate a certain relationship between lysozyme and molting [34]. Lysozymes may directly participate in the insect molting process, or activate the immune mechanism during the molting process, so that the overexpression of immune effectors such as Cclys2 provides protection. The expression levels of *CcLys2* in the second- and fifth-instar nymphs were higher than those at other developmental stages. In the whole life cycle of *C. chinensis*, the second-instar nymph is the only stage with a red body wall, while the fifth-instar nymph is the key stage for adult emergence; therefore, we speculate that the high expression of *CcLys2* at these two stages may be related to the formation of body wall. After infection or injury, insect AMPs are expressed usually in the fat body and secreted into hemolymph to protect insects from pathogenic microorganisms [35,36]. Similar to previous studies, the expression levels of *CcLys2* in the fat body are much higher than those in the other tissues, which points that the fat body is an important immune tissue in the anti-infection process of *C. chinensis*.

Since the immune effectors are believed to be up-regulated in response to infection, the expression of *CcLys2* transcripts in response to the bacterial challenge was analyzed. Compared with the *C. chinensis* without bacterial injection, the expression levels of *CcLys2* continued to increase within 60 h after bacterial injection. In general, AMPs would appear in the hemolymph of infected insects about 6–12 h after infection [37]; therefore, bacterial challenge promoted the up-regulation of *CcLys2*, suggesting that CcLys2 is a very effective immune effector in the innate immunity of *C. chinensis*. Previous studies have revealed that lysozymes in *M. domestica*, in the ruminant stomachs of cattle, and in the digestive organs of marine bivalves also have digestive functions [13,14,38,39]. In *D. melanogaster* and *M. domestica*, the c-type lysozyme in the midgut is involved in the digestion of bacteria ingested after feeding rotten fruits and plant materials [4,40]. This type of lysozyme was highly expressed in the intestinal tract and had the best lysis activity at a low pH value. The expression levels of CcLys2 in the midgut of *C. chinensis* were significantly up-regulated after feeding bacteria. Therefore, CcLys2 may have digestive function or perform immune defense against the sudden increase of bacterial flora in the intestine.

As a part of non-specific immune response, lysozymes are directly involved in various immune responses against pathogenic microorganisms. At present, many studies have evaluated the antibacterial activity of lysozymes. Generally, c-type lysozymes can hydrolyze the β-(1–4)-glucosidic bond between N-acetylmuramic acid and N-acetylglucosamine in peptidoglycan polymer to achieve the bactericidal effect, and are the most effective against Gram-positive bacteria. However, most lysozymes cannot hydrolyze peptidoglycan of the *S. aureus* cell wall because of its O-acetylation modification, so the lysozymes do not affect this bacterium [41]. The resistance of Gram-negative bacteria to c-type lysozymes is due to the fact that their peptidoglycan layer is generally surrounded by an outer membrane and the influence of lysozyme inhibitors makes it difficult for c-type lysozymes to hydrolyze peptidoglycan of bacterial cell wall [42,43]. The recombinant CcLys2 protein was expressed in the *E. coli* BL21 (DE3) system to study the enzyme activity. Like most c-type lysozymes, CcLys2 has significant bacteriolytic effects on Gram-positive bacteria such as *M. luteus* but has no inhibitory effect on *S. aureus* and all Gram-negative bacteria tested. Although previous studies have shown that the ability of lysozyme to inhibit the bacterial growth was not related to or dependent on the hydrolytic activity, lysozymes with muramidase activity were more effective antimicrobial proteins [17,44]. In addition, our results also indicated that CcLys2 may be an intestinal lysozyme. The CcLys2 putative pI was 5.09 and the optimal pH value determined by the turbidimetric method was about 6.0. The typical c-type lysozyme is usually an immune-related alkaline enzyme, but this is not absolute. For some lysozymes used for digestion purposes (such as lysozyme in the midgut of *D. melanogaster*), their acidic pI and optimal pH are closely related to the pH of the intestinal cavity [45,46]. Some intestinal lysozymes involved in immune response have a high pI, but the optimal pH is acidic, such as the lysozyme c-lys4 in *H. axyridis* (pI = 8.18, optimal pH = 6.5) [47].

In insects, lysozyme not only has immune and digestive functions but also is regulated by insect development. Lysozyme also seems to play a role in insects molting; therefore, in the next step, we will further study the function of CcLys2 at the molting stage and the effect of CcLys2 on the growth and development of *C. chinensis* with RNAi technology.

## 5. Conclusions

In this study, a lysozyme in *C. chinensis* was identified by heterologous expression. Our results indicated that CcLys2 belongs to an old phylogenetic group of arthropod c-type lysozymes and the H-branch of the c-type lysozyme. CcLys2 can be strongly induced by bacterial infection, is a very effective immune effector in the innate immunity of *C. chinensis*, and may be related to the control of the flora in the intestine. Like typical c-type lysozymes, CcLys2 is an antibacterial protein with muramidase activity, with the best lysis effect on Gram-positive bacteria. We need to further investigate what the actual functions of this immune-related enzyme are during digestion and bacterial infections.

## Figures and Tables

**Figure 1 biology-10-00330-f001:**
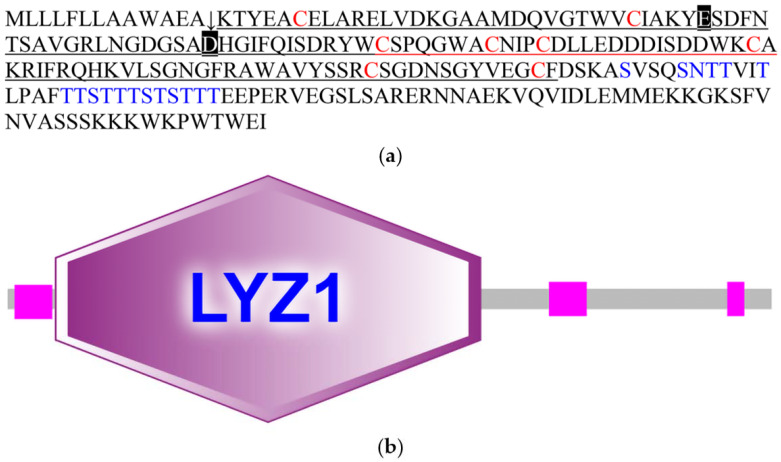
Amino acids and domain structure of CcLys2. (**a**) The amino acid sequence of CcLys2. The arrow denotes the cleavage site of the signal peptide (1–13). The eight cysteine residues are marked in red, E33 and D50 are marked with black boxes, and glycosylation sites are marked in blue. The characteristic domain LYZ1 of the c-type lysozyme is underlined. (**b**) Domain structure of CcLys2. This diagram was generated using SMART and mainly shows the domain LYZ1 of CcLys2.

**Figure 2 biology-10-00330-f002:**
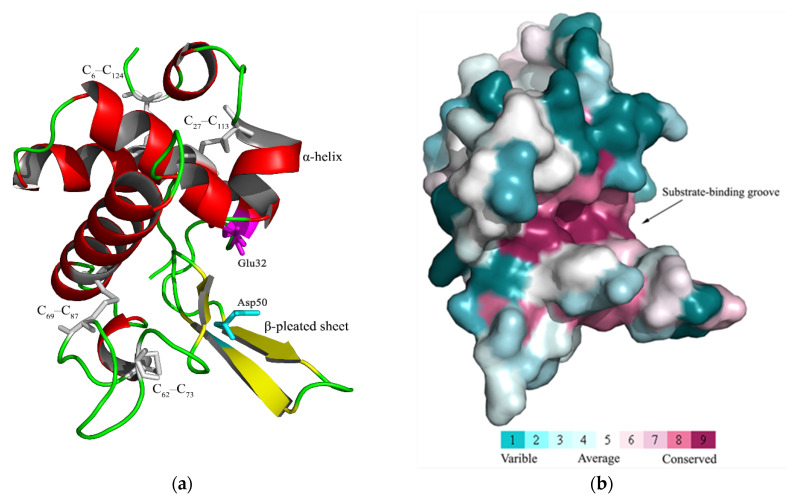
Three-dimensional molecular structure of the mature CcLys2. (**a**) This graphic was generated with PyMOL 2.4 based on the CcLys2.pdb data. C_6_–C_124_, C_27_–C_113_, C_62_–C_73_, and C_69_–C_87_ indicate four disulfide bonds. The putative catalytic site (E32 and D50) is located between the α-helices and β-pleated sheets. (**b**) The highly conserved region (in fuchsin) is displayed in the structure. Homology model was performed by the ConSurf software (https://consurf.tau.ac.il/, 12 February 2021) and optimized using PyMOL 2.4.

**Figure 3 biology-10-00330-f003:**
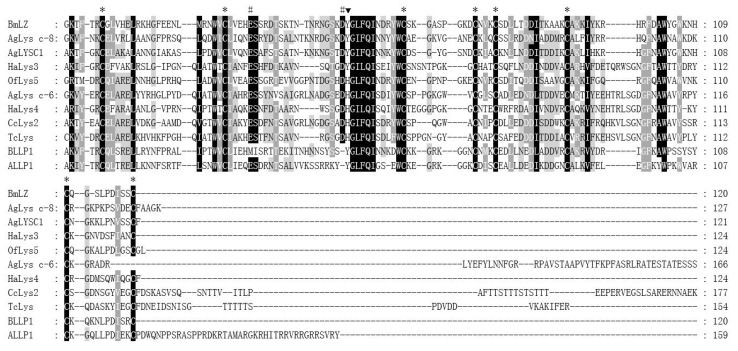
Multiple sequence alignment of CcLys2 with typical c-type lysozymes from 6 other insects. Asterisks mark eight cysteines, pounds mark characteristic catalytic residues glutamate (E) and aspartate (D), and the black triangle marks the characteristic H that replace the conserved Y in the majority of c-type lysozymes. Identical residues are shown in black boxes and strongly conserved residues are shown in gray boxes. Species that lysozymes originate from and GenBank accession numbers of the lysozymes are listed in Appendix A.

**Figure 4 biology-10-00330-f004:**
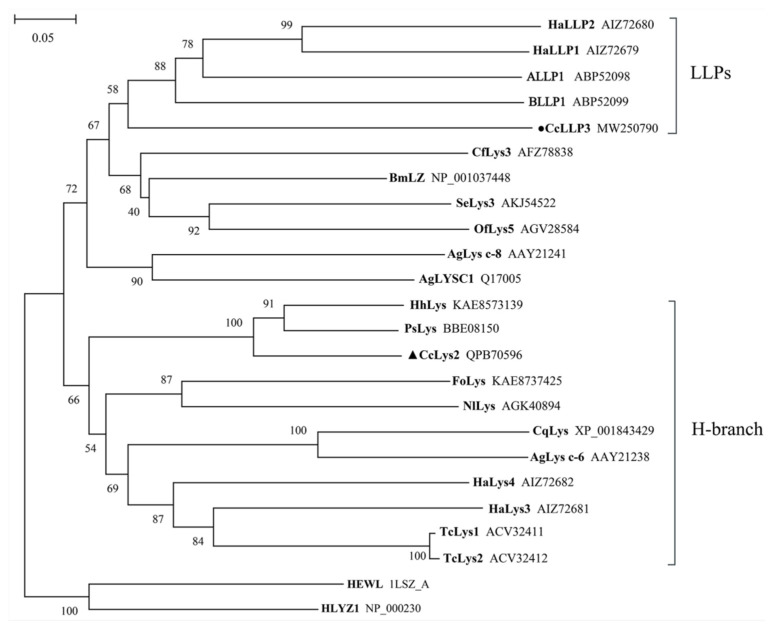
A phylogenetic tree constructed from 19 lysozymes and 5 lysozyme-like proteins (LLPs). This tree was constructed using the neighbor-joining method (NJ) in MEGA X. In total, 1000 replicates were performed and bootstrap confidence values are shown at the node in this tree. The black triangle marks lysozyme from *C. chinensis* and the black dot marks lysozyme-like protein from *C. chinensis*. HLYZ1 of human and HEWL of chicken are used as outgroups. Species that lysozymes originate from and GenBank accession numbers of the lysozymes are listed in Appendix A.

**Figure 5 biology-10-00330-f005:**
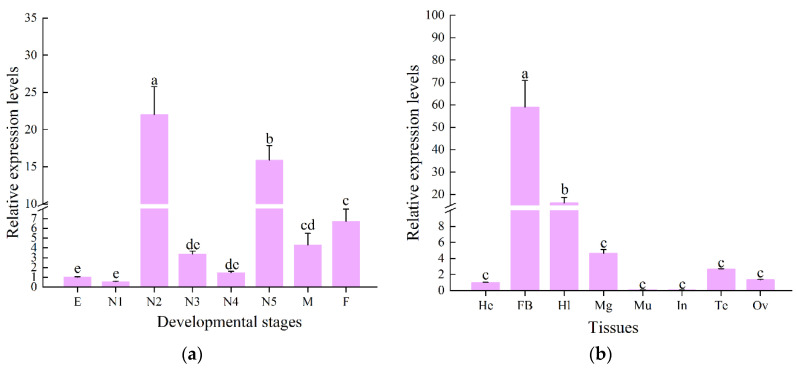
Spatiotemporal expression profile of *CcLys2*. (**a**) Expression levels of *CcLys2* at various developmental stages. E: egg; N1–N5: first–fifth-instar nymphs; F: female; M: male. (**b**) Expression levels of *CcLys2* in different adult tissues. He: head; FB: fat body; Hl: hemolymph; Mg: midgut; Mu: muscle; In: integument; Te: testis; Ov: ovary. Data are expressed as the mean ± SD. Different letters above the bars represent significant differences at *p* < 0.05 based on Duncan’s test.

**Figure 6 biology-10-00330-f006:**
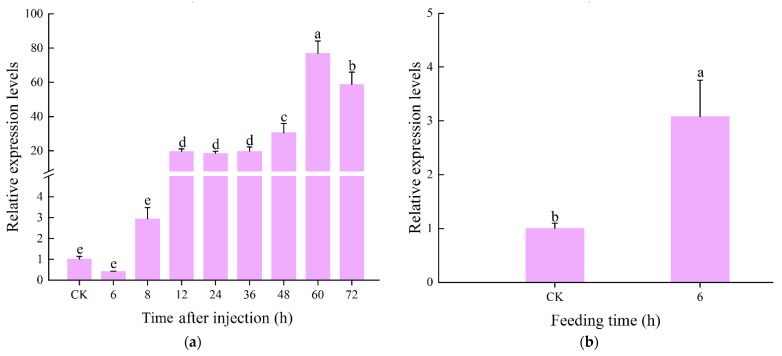
Relative expression levels of *CcLys2* at different times of induction. (**a**) Relative expression levels of *CcLys2* at different time after injecting bacteria. (**b**) Relative expression levels of *CcLys2* in the midgut 6 h after feeding bacteria. CK denotes the blank control group without any treatment. Data are expressed as the mean ± SD. Different letters above bars represent significant differences at *p* < 0.05 based on Duncan’s test.

**Figure 7 biology-10-00330-f007:**
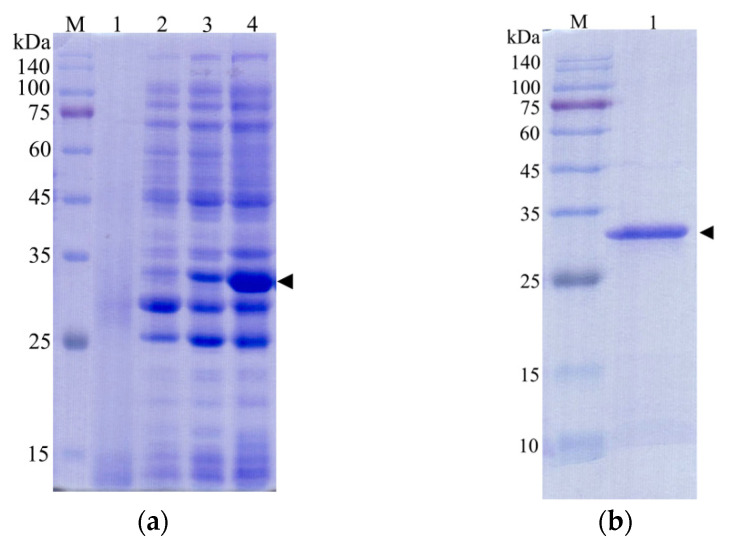
Identification of recombinant CcLys2 protein. (**a**) SDS-PAGE analysis of the bacterial lysate. Lane 1: the non-induced bacterial culture; Lane 2: the soluble supernatant; Lanes 3 and 4: inclusion body protein. (**b**) Purified CcLys2. Lane 1: fusion CcLys2. (**c**) Western blot analysis of CcLys2. Lane 1: blotting band of fusion CcLys2. M: Protein molecular weight marker (10–140 kDa). The arrow indicates the bands of fusion CcLys2.

**Figure 8 biology-10-00330-f008:**
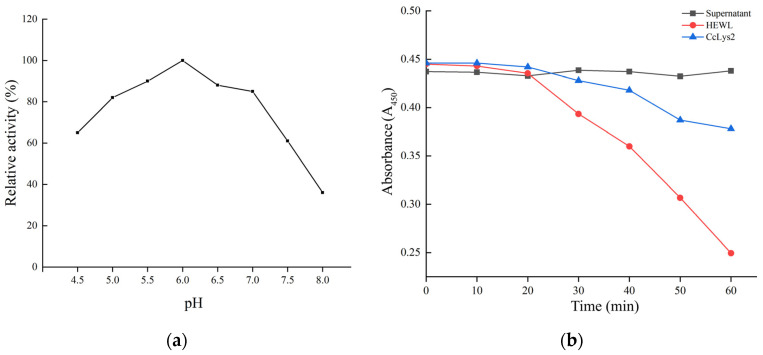
Optimum pH and muramidase activity of CcLys2. (**a**) The relative activity of CcLys2 was determined using the turbidimetric method at different pH values (4.5–8.0). (**b**) CcLys2, HEWL, and the non-induced bacterial culture (supernatant) were incubated with *M. luteus* and the change in OD_450_ over time was recorded.

**Figure 9 biology-10-00330-f009:**
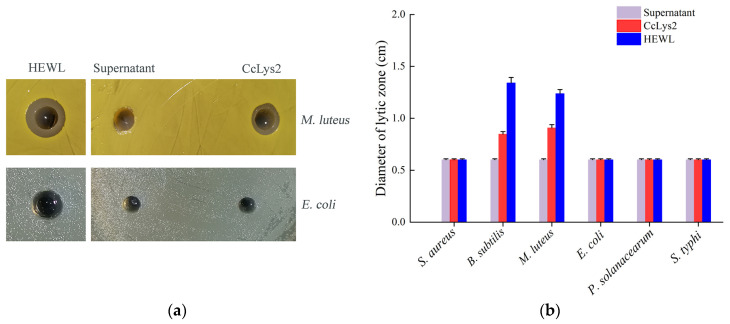
Antibacterial activity of CcLys2. (**a**) The CcLys2 lytic zone of *Micrococcus luteus* and *Escherichia coli* were observed at pH 6.0. (**b**) Lytic zones produced by CcLys2 were measured against the six bacteria. For the plate without a lytic zone, the diameter of a hole was used to represent the one of a lytic zone (0.6 ± 0.01 cm). Data are expressed as the mean values with standard deviations.

**Table 1 biology-10-00330-t001:** Primers used for verification and expression analyses of *CcLys2*.

Primer Name	Sequence	Primer Usage
T7-F	5′-TAATACGACTCACTATAGG-3′	Clone
T7-R	5′-GCTAGTTATTGCTCAGCGG-3′	
Lys-qF	5′-CTCTTGGAGGACGACGACATCT-3′	RT-qPCR
Lys-qR	5′-TGACTGTGGTGTTGGACTGTGA-3′	
Actin-F	5′-ACCGCTGAGAGGGAAATCG-3′	
Actin-R	5′-CAAGAAGGAAGGCTGGAAGAG-3′	

## Data Availability

The data presented in this study are available on request from the corresponding author.

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
