# Peer review of "Identification and Functional Analysis of a Lysozyme Gene from Coridius chinensis (Hemiptera: Dinidoridae)"

_biology, 2021, doi:10.3390/biology10040330_

Round 1

Reviewer 1 Report

Huang et. al present their work, entitled, “Identification and  functional analysis of a lysozyme gene from Coridius chinensis (Hemiptera: Dinidoridae)”. In this study, the authors identify a lysozyme, CcLys2, from C. chinensis. Excitingly, CcLys2 was significantly up-regulated after injecting and feeding bacteria. Furthermore, in a growth inhibition test of bacteria, the authors find that CcLys2 has antibacterial activity against Gram-positive bacteria at a low pH. This demonstrates a novel immune role of the lysozyme in Coridius chinensis innate immunity and perhaps as a potential therapeutic molecule.

I think that this is an extremely thorough body of work from the gene level to the transcriptional level, to the organismal level. I congratulate the authors for this great work.

Major Comments

The authors provide an important body of work furthering the knowledge of CcLys2 levels and its connection to bacteria (Figures 5, 6, 9). My only suggestion is genetic knockdown of CcLys2 and observing whether survival is affected. My prediction would be that animals with the gene knocked down would be unable to survive infection. Of course, this may not be possible given the organism you are working with, but injection of the synthetic dsRNA should overcome this obstacle. As a control, a dsRNA against GFP should be performed (to induce RNAi machinery). I think this final experiment would show the necessity and sufficiency of this extremely important lysozyme. I do see that the authors are prosing using RNAi in the context of molting, but I believe using the technique here would further solidify the observation and prove most valuable to the finding and impact.

Minor Comments

I believe that a summary figure would be most beneficial for the manuscript. There is a great deal of high quality data that can be condensed to an easily interpretable graphic.

There are some general sentence structure/typing issues to resolve. I listed a few below:

  • Line 43: change “as it can” to “and was found to”
  • Line 56-57: sentence is missing subject-verb agreement
  • Line 60: sentence is difficult to understand
  • Line 426: “highest” typo

Reviewer 2 Report

MDPI Biology: Identification and functional analysis of a lysozyme gene from Coridius chinensis (Hemiptera: Dinidoridae)

I enjoyed reading this professionally written and detailed manuscript on CcLys2, which belongs to arthropod c-type lysozymes. The authors show that bacterial infections can induce CcLys2, and, as a part of innate immunity, it is dedicated to controlling the intestinal microbiome. I only suggest some more polishing of the language.

For example, lines 26-28: The sentence “In order to search for the effective medicinal components in the extract of C. chinensis and clarify the functions of the lysozyme in C. chinensis” requires extension and further explanation. "In order" can be omitted.

Line 697: Delete “Of course,” as redundant.

The authors might also be interested in shortening other parts of the manuscript to increase the paper's overall focus.

Author Response

Point 1: For example, lines 26-28: The sentence “In order to search for the effective medicinal components in the extract of C. chinensis and clarify the functions of the lysozyme in C. chinensis” requires extension and further explanation. "In order" can be omitted.

Line 697: Delete “Of course,” as redundant.

Response 1:  We have made some modifications that are marked in red in this manuscript, according to the reviewer’s comments. Please see the manuscript with revision marks.

  • Page 1, line 12: "In order" have been deleted.
  • Page 1, lines 26-28: “In order to search for the effective medicinal components in the extract of chinensisand clarify the functions of the lysozyme in C. chinensis” is modified into “Studying lysozyme in C. chinensis will be helpful to further explore the evolutionary relationship and functional differences among lysozymes of various species and to determine whether they have biological activity and medicinal value”.
  • Line 729: “Of course,” have been deleted.

Point 2: The authors might also be interested in shortening other parts of the manuscript to increase the paper's overall focus.

Response 2: We have revised the grammar of the manuscript and deleted the redundant parts in the article to increase the paper's overall focus. Please see the manuscript with revision marks.

Round 2

Reviewer 1 Report

The authors have addressed my concerns. I appreciate the new figures and added explanation of experimental approaches in the manuscript. It is my recommendation that this important manuscript be accepted.

Balint Z. Kacsoh | Ph.D.
Berger Lab
Epigenetics Institute
Perelman School of Medicine at the University of Pennsylvania
Philadelphia, PA 19104
website: www.balintzkacsoh.com
email: balintzkacsoh@gmail.com